# The Role of Berry Consumption on Blood Pressure Regulation and Hypertension: An Overview of the Clinical Evidence

**DOI:** 10.3390/nu14132701

**Published:** 2022-06-29

**Authors:** Stefano Vendrame, Tolu Esther Adekeye, Dorothy Klimis-Zacas

**Affiliations:** School of Food and Agriculture, University of Maine, Orono, ME 04469, USA; stefano.vendrame@fulbrightmail.org (S.V.); tolu.adekeye@maine.edu (T.E.A.)

**Keywords:** berries, blood pressure, anthocyanins

## Abstract

The existence of a relationship between the consumption of dietary berries and blood pressure reduction in humans has been repeatedly hypothesized and documented by an increasing body of epidemiological and clinical evidence that has accumulated in recent years. However, results are mixed and complicated by a number of potentially confounding factors. The objective of this article is to review and summarize the available clinical evidence examining the effects of berry consumption on blood pressure regulation as well as the prevention or treatment of hypertension in humans, providing an overview of the potential contribution of distinctive berry polyphenols (anthocyanins, condensed tannins and ellagic acid), and results of dietary interventions with blueberries, bilberries, cranberries, raspberries, strawberries, chokeberries, cherries, blackcurrants and açai berries. We conclude that, while there is insufficient evidence supporting the existence of a direct blood pressure lowering effect, there is stronger evidence for specific types of berries acting indirectly to normalize blood pressure in subjects that are already hypertensive.

## 1. Introduction

It is widely recognized that diet can significantly affect blood pressure (BP), and several dietary patterns have been associated with positive or negative effects on BP regulation both in the short and the long term [1].

Among these patterns, regular consumption of fruits and vegetables is an established protective factor against hypertension, due to a combination of factors including a high nutrient but low energy density, which helps maintain a healthy weight; a high content of potassium paired with low content of sodium, with a consequent diuretic effect [2]; the contribution to calcium [3] and magnesium requirements, both necessary for correct BP regulation [4]; the fiber content, which helps attenuate glucose and insulin peaks [5]; and displacement of other food items whose excess consumption is detrimental to BP due to their unfavorable lipid profile or high caloric density [6].

Of particular importance is also the abundance of vitamin C. A systematic review and meta-analysis of 29 randomized controlled clinical trials examining the effects of vitamin C intake on BP, found a statistically significant pooled effect of vitamin C in lowering both systolic blood pressure (SBP) and diastolic blood pressure (DBP) in all participants, and a significantly larger reduction in SBP when only trials with hypertensive subjects were considered [7].

However, while this nutritional profile is common to most fruits and vegetables, the specific effect proposed for berries on BP must involve further features, acting in addition or synergistically to the above mentioned ones. What is distinctive to berries is the presence of a number of phytochemicals within the family of polyphenols [8].

## 2. Berry Polyphenols and Blood Pressure

The term polyphenols indicates a large class of naturally occurring plant secondary metabolites, including four principal classes: phenolic acids, flavonoids, stilbenes, and lignans [9]. Many of these molecules have known antioxidant, anti-inflammatory, anti-thrombotic and vascular protective effects that make them strongly protective against cardiovascular diseases [9]. Among their therapeutic potential, BP regulation has been repeatedly cited, including prevention as well as treatment of hypertension [10].

Within berries, there are three classes of polyphenols in particular that may account for a beneficial effect on BP: anthocyanins, condensed tannins and ellagic acid [8].

### 2.1. Anthocyanins

Anthocyanins (ACN) are water-soluble pigments of the flavonoid family of polyphenols, made of a sugar covalently bound to a class of highly reactive molecules with a flavylium cation structure, called anthocyanidins [11].

Chokeberries and black raspberries are major sources of ACN (>400 mg/100 g), followed by blackcurrants, bilberries, blueberries and blackberries (over 100 mg/100 g), red raspberries and cherries (over 50 mg/100 g). Cranberries and strawberries have lower ACN contents (less than 50 mg/100 g) and contain higher amounts of other bioactive phytochemicals [12].

A consistent body of epidemiological and clinical evidence suggests the existence of an inverse relationship between ACN intake and BP, likely due to multiple mechanisms that have been repeatedly demonstrated in vitro and in animal studies [13] and are schematized in Figure 1. The major mechanism involves the ability of ACN to increase endothelial-derived nitric oxide (NO), thus enhancing endothelium-dependent vasorelaxation and preventing calcium-induced vascular smooth muscle contraction. This effect is exerted via the enhancement of endothelial NO synthase (eNOS) expression and activity, which in turn activates soluble guanylate cyclase, increasing cGMP, blocking the release of intracellular calcium [14]. Moreover, the strong antioxidant activity of ACN results in prevention of oxidative damage, and especially radical-induced NO conversion mediated by NADPH oxidase, which has a known detrimental effect on endothelial function [15]. Finally, ACN have been shown to attenuate the synthesis of several molecules with known vasoconstricting effects such as endothelin-1 (ET-1), thromboxanes (via cyclooxygenase (COX) pathway inhibition) and angiotensin II (via angiotensin-converting enzyme (ACE) activity inhibition) [16].

Analyzing pooled data on over 155,000 men and women from three large prospective epidemiological studies (the Nurses’ Health Study, the NHS I and the Health Professionals Follow-Up Study), with over 34,000 cases of hypertension developed during a 14 year follow-up, the association between flavonoid intake and incident hypertension was investigated [17]. Of all classes of flavonoids, only ACN intake was associated with lower hypertension risk, with subjects in the highest ACN intake quintile having an 8% risk reduction compared to subjects in the lowest quintile. The authors also noted that most of the ACN intake in the population came from consumption of blueberries and strawberries. When individual compounds were analyzed, instead of whole classes, a slightly smaller but significant risk reduction was also found in association with the intake of apigenin (a flavone), and of catechin (a flavan-3-ol) [17].

Similar results were found in a cross-sectional study analyzing data from a cohort of about 1900 adult women from the TwinsUK registry: a significant reduction in central SBP and mean arterial pressure (MAP) was associated with a higher intake of ACN, but not with total flavonoid intake or with any of the other flavonoid subclasses [18].

However, the clinical evidence is weaker: although a few individual clinical trials have reported a BP lowering effect following administration of isolated ACN [19], three meta-analyses of clinical trials investigating the effects of ACN supplementation documented that the reduction in SBP or DBP did not reach statistical significance [20,21,22].

### 2.2. Condensed Tannins (Proanthocyanidins)

Condensed tannins, or proanthocyanidins, are polymeric compounds of the flavonoid family, with various hydroxylation and length unique to their functions. They are formed by the polymerization of flavan-3-ols and can yield anthocyanidins upon hydrolysis in acidic environment [23].

Blueberries, cranberries and chokeberries are major sources of proanthocyanidins, followed by blackcurrants and strawberries [24].

Several studies have explored the therapeutic effects and mechanisms of proanthocyanidins on BP. A randomized study reported a significant decrease in SBP and DBP in pre-hypertensive subjects after consuming 400 mg of grape seed proanthocyanidin extract daily for twelve weeks. This study reported that vascular stiffness was abolished by proanthocyanidin tablets in non-smoking pre-hypertensive subjects [25].

Another study reported that BP decreased after four weeks of treatment with 100 mg/day and 200 mg/day of grape seed proanthocyanidin in middle aged menopausal women [26]. In a five-week randomized study, 150 mg/day of a proanthocyanidin extract, decreased SBP and oxidized LDL (oxLDL), with increased HDL in stage-1 hypertensive men [27]. Similarly, a controlled registry study reported normalized BP in 93% of pre- and mild hypertensive subjects with decreased antioxidative parameters after four months of 300 mg/day proanthocyanidin extract [28].

An important mechanism proposed for the activity of condensed tannins on BP involves inhibition of the ACE pathway [29].

Other mechanisms of proanthocyanidins resulting in BP regulation in humans may involve antioxidative scavenging of oxLDL and LDL-cholesterol [30] and removal of carotid atherosclerosis plaque [31].

Other anti-hypertensive, vasoregulatory and vasoprotective mechanisms of proanthocyanidins have been reported in animal and *in vitro* studies [32].

Proanthocyanidin treatment ameliorated oxidative stress and protected the blood brain barrier during arteriosclerosis by inhibiting oxLDL dock to its receptor LOX-1 to prevent cerebrovascular diseases [33]. Additionally, 0.1% and 1% rich proanthocyanidin extract incorporated into rabbit diets attenuated cholesterol-induced aortic lesions and atherosclerosis [34]. The same study reported decreased oxLDL activation and foam cells via antioxidative mechanism [34]. An *in vitro* study reported improved cell contractility, survival and antioxidation in chick cardiomyocytes pretreated with grape seed proanthocyanidin extract before antimycin A-induced oxidative stress [35]. Similarly, a long-term administration of proanthocyanidin extract from grape seed provided an antihypertensive effect in male rats with established cafeteria-induced hypertension. Three weeks of proanthocyanidin treatment significantly decreased both the systolic and diastolic BP with upregulated vasoregulatory mechanisms such as Sirt-1 gene, and downregulated vasoconstrictor ET-1 [36]. Grape seed proanthocyanidin extract, administered orally, attenuated aortic stiffness caused by ouabain by decreasing both ET-1 and tumor growth factor-β1 in thoracic aorta of Sprague-Dawley rats after five weeks of treatment [37].

### 2.3. Ellagic Acid

Ellagic acid is a polyphenol, whose structure is that of a dimeric derivative of gallic acid, although its formation in plants is mostly from the hydrolysis of larger, polymeric phenolics called ellagitannins [38]. Strawberries, raspberries, and blackberries are major sources of ellagic acid (over 30 mg/100 g) and ellagitannins.

Although data from human studies are scarce and mostly indirect, there have been several studies with animal models reporting beneficial effects of ellagic acid on BP and cardiovascular health [39]. Intravenous administration of 5 mg/kg ellagic acid decreased BP after 20–30 sec in Wistar rats via a vasodilatory mechanism similar to histamine liberators such as dextran [40]. Similarly, a ten-day study reported that ellagic acid pretreatment significantly decreased arterial BP and heart rate with antioxidative effect in isoproterenol-induced hypertensive male Wistar rats. The 15 mg/kg ellagic acid pretreatment, reduced heart rate and BP comparable to the normal non-hypertensive group [41]. In comparison with the group not treated with isoproterenol, cardiac markers such as troponin-I and C-reactive proteins were significantly decreased. The same study reported that antioxidant markers, such as superoxide dismutase, glutathione peroxidase and catalase, were significantly increased in heart tissues of the 15 mg/kg ellagic acid pretreatment group compared with the non-treated isoproterenol group, whereas lipid peroxidation was decreased.

Orally administered 15 mg/kg ellagic acid for five weeks significantly decreased BP and oxidative stress in NG-Nitro-L-arginine methyl ester (L-NAME) induced hypertensive rats. This treatment partially restored NO production and endothelial nitric oxide synthase activities to ameliorate hypertension [42]. Similarly, ellagic acid prevented monocrotaline-induced pulmonary hypertension in male Sprague–Dawley rats with daily treatment of 30 mg/kg and 50 mg/kg after four weeks. The treatment prevented inflammasome activation and suppressed oxidative species in the pulmonary vascular region [43]. Wistar rats fed a high carbohydrate and fat diet for 16 weeks with ellagic acid (0.8/kg of diet) showed decreased SBP and left ventricular weight, as well as improved ejection fraction, fractional shortening, and vascular function [44]. An *in vitro* human ACE study observed anti-hypertensive effects (among other metabolites) with ellagic acid treatment similar to ACE inhibitor captopril: the ellagic acid treatment had a comparable docking interaction on the ACE active site similar to captopril in an *in-silico* study [45]. Hypertensive rats induced with L-NAME hydrochloride showed significant decrease in BP after five weeks of 10 and 30 mg/kg ellagic treatment with improved levels of serum nitrite/nitrate bioavailability essential for vasoregulation [46]. Oral ellagic acid treatment normalized BP and improved mitochondrial functions in isoproterenol induced hypertensive rats via β-adrenergic antagonist mechanism to ameliorate cardiac infarction and promote vascular health [47].

## 3. Berry Consumption and Blood Pressure

As far as clinical evidence regarding berry consumption and BP is concerned, when a large meta-analysis was performed, pooling data from 128 clinical trials examining different sources of ACN, for a total of over 5500 participants, a significant reduction in both SBP and DBP was found with consumption of berries, as well as red grapes/red wine [48].

Such results are partially confirmed by a previous, smaller meta-analysis focusing specifically on berries (22 clinical trials, 1251 subjects), which also reported a significant effect of berry consumption in reducing SBP, but not DPB [49].

A large double-blind, placebo-controlled, parallel arms study was conducted specifically investigating the effect of a mix of berries on BP. A group of 134 pre-hypertensive or hypertensive participants received either a daily serving of 500 mL juice made of red grape, chokeberry, cherry and bilberry (43 mg total ACN), the same juice enriched with blackcurrant press residue (210 mg total ACN), or a placebo. After 12 weeks of treatment, a statistically significant reduction in SBP, but not DBP, was observed in both juice groups. Furthermore, the reduction in BP was more pronounced in the subgroup with higher BP at recruitment [50].

Five other trials investigating the effects of mixed berries recorded BP, although it was not the main outcome and they were not conducted specifically on hypertensive participants. No effects on BP were observed [51,52,53,54], with the exception of a single-blind, placebo-controlled, parallel arms trial in which 8 weeks consumption of a daily mix of berries (bilberries, lingonberries, blackcurrant, strawberry, chokeberry, raspberry) providing 515 mg total ACN, resulted in a significant reduction of SBP, but not DBP, in a group of 72 participants with cardiovascular risk factors [55]. Additionally in this study, the authors noted that the effect on SBP was very strong in the subgroup with higher baseline BP values.

Overall, these data consider the effects of a mix of berries on a mixed population, including men and women of all age groups and from different geographical regions, both healthy and with cardiovascular risk factors. In the following paragraphs, aiming to pinpoint more targeted effects, an overview of the clinical evidence examining the effects of specific berries on specific populations will be provided.

To this aim, scientific literature was searched using the PubMed database (updated April 2022) using the following key: (“blood pressure’’ OR ‘‘SBP’’ OR ‘’DBP’’) AND (“blueberry *’’ OR “cranberry *’’ OR “strawberry *’’ OR “bilberry *’’ OR “raspberry *’’ OR “chokeberry *’’ OR “cherry *’’ OR “blackcurrant’’ OR “açai”) and restricting the search to clinical trials and randomized controlled trials published over the last 20 years in the English language.

A total of 110 records were identified and screened for relevance. Upon removal of duplicates or ineligible studies, a total of 86 papers were analyzed for extraction of the following data: type of berry, study design, number and characteristics of participants, treatment duration and dose (with ACN content where available), and findings restricted to BP outcomes. A summary of results can be found in Table 1.

### 3.1. Family Ericaceae

To the family Ericaceae belong berries of the genus *Vaccinium*, including wild blueberries (*V. angustifolium*), highbush blueberries (*V. corymbosum*), bilberries (*V. myrtillus*), cranberries (*V. macrocarpon*) and lingonberries (*V. vitis-idaea*).

While the presence of condensed tannins characterizes all berries within this family, blueberries and bilberries are primarily a source of ACN, while cranberries contain lower concentrations of ACN but a larger pool of other polyphenols, in particular flavonols such as quercetin [12].

#### 3.1.1. Highbush and Low Bush (Wild) Blueberry

The effect of both single-dose and longer term blueberry consumption on BP was investigated in numerous dietary interventions, with interesting but mixed results.

A single serving of highbush blueberry drink providing 384 mg ACN did not have any effect on SBP or DBP in a group of 10 young adults, but the same single serving was able to significantly decrease SBP and improve endothelial function in a group of 16 young smokers, after one cigarette, with no effect on DBP [56]. However, in a subsequent experiment on a group of 12 young males, a single serving of blueberry drink providing 309 of ACN did not affect SBP or DBP and did not restore BP after smoking a cigarette, although endothelial function was improved [57].

A single serving of a highbush blueberry drink providing 310 mg of ACN, while improving endothelial function, did not affect SBP or DBP in a group of 10 healthy young male participants [58]. In a subsequent experiment with 10 healthy subjects, a single serving of freeze-dried blueberry drink providing 340 mg of ACN, or a single serving of a blueberry baked product made with the same amount of blueberry powder, both improved endothelial function but did not affect SBP or DBP [59].

A significant decrease in both SBP and DBP was observed in a double-blind, placebo-controlled, parallel arms intervention, in which 48 postmenopausal women with pre- and stage-1 hypertension received for 8 weeks a daily serving of 480 mL highbush blueberry drink (made with 22 g freeze-dried berry powder and equivalent to one cup of fresh berries) providing 469 mg of ACN [60].

In a single-blind, water-controlled, parallel arms trial, 48 obese participants with metabolic syndrome received a daily serving of highbush blueberry drink made from 50 g freeze-dried blueberry powder (equivalent to 350 g fresh berries) providing 742 mg ACN. After eight weeks, although no effects were observed in lipid profile or inflammatory markers, both SBP and DBP were significantly lower [61].

In a placebo-controlled, parallel arms trial, a group of 25 middle-aged participants received a daily serving of freeze-dried highbush blueberry powder equivalent to a cup of fresh berries, for six weeks. A significant reduction in aortic systolic pressures (ASPs) and SBP, but not DBP, was recorded. Furthermore, when only the subset of pre-hypertensive subjects was considered, DBP was also significantly decreased [62].

In a controlled, parallel-arm trial, a group of 34 women in early pregnancy with previous history of gestational diabetes, received a daily serving of 280 g whole frozen highbush blueberries (700 mg ACN) and a 12 g soluble fiber supplement for 18 weeks. A significant reduction in DBP, but not SBP, was observed, together with other positive outcomes in blood glucose control and inflammation [63].

In a double-blind, placebo-controlled, parallel arms intervention, in which a daily serving of 40 g freeze-dried highbush blueberry powder was given to 63 patients with knee osteoarthritis for 4 months, SBP was significantly lowered, although no effect was observed for DBP [64].

In another double-blind, placebo-controlled, parallel arms trial, 122 older adults were enrolled and given for six months a daily serving of either one or two grams of whole wild blueberry powder, or 200 mg of a wild blueberry extract, providing 2.7, 5.4 or 14 mg of ACN, respectively. Interestingly, a significant reduction of SBP was observed with the extract, but not with any of the whole powders, suggesting the existence of a dose dependency [65]. Furthermore, the BP lowering effect of the wild blueberry extract was already significant after 3 months, halfway through the intervention period. It must also be noted that the highest dose of wild blueberry used in this study is still much lower compared to the amount of blueberries used in all the other studies [65].

Conversely, in a double-blind, placebo-controlled, parallel arms trial, a group of 115 obese or overweight subjects with metabolic syndrome received a daily serving of either 13 or 26 mg freeze-dried highbush blueberry powder (equivalent to half a cup or one cup of fresh berries) providing 182 mg or 364 mg ACN, for six months. While markers of endothelial function improved, no effect was observed on SBP or DBP [66].

A cup of fresh highbush blueberries given daily for 3 weeks to a group of 20 young smokers, in a controlled, parallel arms trial, did not affect SBP, DBP or activity of the ACE [67].

In a double-blind, placebo-controlled, parallel arms intervention, six weeks consumption of a daily serving of 45 g highbush blueberry powder (equivalent to two cups of fresh berries) providing 580 mg of ACN, did not affect SBP or DBP in a group of 32 obese participants with insulin resistance [68]. In a subsequent experiment with the same experimental design and blueberry intake, no effect on SBP or DBP was observed in a group of 44 obese participants with metabolic syndrome [69].

In a placebo-controlled, crossover trial with 18 male participants with cardiovascular risk factors, six weeks consumption of a daily wild blueberry drink (made of 25 g freeze-dried berry powder) providing 400 mg ACN, did not result in any effect on SBP or DBP [70].

In a single-blind, placebo-controlled, crossover trial, no effect on SBP or DBP was observed in a group of 19 women at risk for type II diabetes, following consumption of a daily 240 mL serving of wild blueberry juice for one week [71].

Finally, a meta-analysis of six clinical trials available as of 2016, for a total of 204 participants, was not able to find a statistically significant effect of blueberry consumption on BP [137].

In conclusion, only one out of six single-dose trials, but six out of twelve interventions with longer term blueberry consumption, were able to detect a BP lowering effect. Only one of these studies specifically targeted the effects on BP in hypertensive participants and found a positive effect for blueberry consumption. Two other studies also suggest that the effect on BP is more evident when baseline values are higher, for example when only the subset of hypertensive participants is considered, or when participants are asked to smoke a cigarette.

#### 3.1.2. Bilberry

Three human interventions investigated the effect of bilberry consumption on BP.

Daily administration of a 1.4 g dose of bilberry extract for four weeks to a group of 20 participants with T2DM, in a double-blind, placebo-controlled, crossover intervention, did not significantly affect SBP or DBP [72].

In a controlled, parallel arms intervention with 27 overweight or obese subjects with metabolic syndrome, daily consumption of a 200 g bilberry puree plus 40 g dried bilberries (equivalent to a total of 400 g fresh berries), had no effect on SBP or DBP [73].

A pre-post intervention with a serving of frozen bilberries providing 456 mg of ACN, given three times per week to a group of 36 healthy subjects for a period of six weeks, did not have any effect on SBP or DBP [74].

Overall, none of the available studies was able to find an effect of bilberry consumption on BP, although no study specifically targeted BP or hypertensive subjects.

#### 3.1.3. Cranberry

In a controlled, crossover trial, s single dose of 40 g dried cranberry powder administered following a high fat meal to a group of 40 obese participants with type II diabetes, did not affect SBP or DBP after 1, 2 or 4 h [75].

A single dose of 480 mL cranberry juice following a meal, improved endothelial function but did not affect SBP or DBP in a group of 15 overweight or obese participants with coronary artery disease [76]. The same dose of cranberry juice given for 4 weeks to a group of 44 overweight or obese participants with coronary artery disease in a placebo-controlled, crossover trial, also improved endothelial function without affecting SBP or DBP [76].

Similarly, in a double-blind, placebo-controlled, parallel arms intervention, a daily serving of 480 mL cranberry juice for 8 weeks, improved endothelial function but did not affect SBP or DBP in a group of 31 women with metabolic syndrome [77].

In a double-blind, placebo-controlled, parallel arms trial, no effect on SBP or DBP was observed in a group of 35 participants with obesity and impaired glucose tolerance, receiving a daily serving of 450 mL low-energy cranberry beverage for 8 weeks [78].

In another double-blind, placebo-controlled, parallel arms intervention, daily intake of a 500 mg capsule of cranberry powder extract for 12 weeks, did not affect SBP or DBP in a group of 30 subjects with type II diabetes [79].

In a placebo-controlled, parallel arms study, 55 middle-aged participants received a twice daily serving of cranberry juice providing either 62 or 173 mg of total phenolics. After 8 weeks of intervention, a significant reduction in DBP, but not SBP, was measured [80].

In a placebo-controlled, crossover intervention, 40 overweight or obese participants with pre-hypertension, received a daily serving of 500 mL cranberry drink (27% cranberry juice) for 8 weeks. Although no change in SBP or DBP was observed, 24-h ambulatory BP measurement revealed a significant reduction in DBP during daytime hours [81].

In a double-blind, placebo-controlled, crossover trial, a daily serving of 500 mL cranberry drink (27% cranberry juice) for 4 weeks did not affect SBP or DBP in a group of 35 abdominally obese men with or without metabolic syndrome [82].

In a subsequent experiment, the dose was adjusted to body weight to obtain servings of 7 mL cranberry juice for each kg of body weight, given daily for 2 weeks to a group of 21 abdominally obese men with dyslipidemia. No effect on SBP or DBP was observed [83].

In yet another experiment, the effect of increasing dosages was investigated in a group of 30 abdominally obese men with or without metabolic syndrome, who received daily servings of 125 mL cranberry juice for 4 weeks, followed by 250 mL/day for 4 weeks and 500 mL/day for 4 weeks, for a total of 12 weeks of intervention. No effect on DBP was observed, but a significant reduction in SBP was recorded following consumption of the highest dose, suggesting either a dose-dependent (the effect was not observed with 125 or 250 mL/day servings) or a time-dependent effect (the effect was not observed after 4 or 8 weeks of intervention) [84].

In conclusion, only three out of eleven studies reported a blood-pressure lowering effect associated with cranberry consumption, while eight studies reported no effect. However, only one study specifically targeted pre-hypertensive subjects, reporting a significant effect on DBP.

Overall, the available evidence linking cranberry consumption and BP control appears to be extremely limited and inconclusive.

### 3.2. Family Rosaceae

To the family Rosaceae belong strawberries (*Fragaria spp.*), chokeberries (*Aronia melanocarpa*), sweet cherries (*Prunus avium*), sour cherries (*Prunus cerasus*) and berries of the genus *Rubus*, including red raspberries (*Rubus idaeus*), black raspberries (*Rubus occidentalis*) and blackberries (various *Rubus* spp.).

Chokeberries are among the berries with the highest total phenolic content (around 2% in weight). Content of ACN is highest in chokeberries and black raspberries, and lower in red raspberries, cherries and strawberries. Strawberries and *Rubus* spp. berries (raspberries and blackberries) are unique among berries in having a high content of ellagic acid and ellagitannins. Chokeberries and strawberries are also very good sources of condensed tannins [12].

#### 3.2.1. Red and Black Raspberry

Post-prandial assessment with or without a single serving of frozen red raspberries (225 mg ACN) revealed no effect on SBP or DBP in a group of 25 obese participants with type II diabetes [85].

However, when the same serving of red raspberries was given daily for a period of 4 weeks to a group of 22 obese subjects, in a controlled, crossover trial, a statistically significant reduction of SBP, but not DBP, was measured [85].

In a controlled, parallel arms intervention, daily servings of 280 g frozen red raspberries for 8 weeks, had no effect on SBP or DBP in a group of 59 overweight or abdominally obese subjects with slight hyperinsulinemia or hypertriglyceridemia [86].

In a double-blind, placebo-controlled, parallel arms intervention on 41 patients with metabolic syndrome, a daily serving of 750 mg dried unripe black raspberry powder for 12 weeks had no effect on SBP or DBP [87]. In a subsequent double-blind, placebo-controlled, parallel arms experiment, two higher doses (1500 mg or 2500 mg) of the same powder were given daily for 8 weeks to a group of 45 participants with pre-hypertension. After 8 weeks, a significant reduction in SBP, but not DBP, was measured but only in the group receiving the highest dose of unripe black raspberry powder [88].

In conclusion, two out of five studies reported an effect of raspberry consumption on BP. Only one study directly investigated pre-hypertensive subjects, reporting a significant BP lowering effect, but only following consumption of the highest dose tested.

#### 3.2.2. Strawberry

In a placebo-controlled, crossover trial, a single dose of 40 g freeze-dried strawberry powder, equivalent to 450 g fresh strawberries, given to 30 overweight or obese individuals following a high-fat meal, did not alter post-prandial SBP [89].

Conversely, in a double-blind, controlled, parallel arms trial, a significant reduction in SBP was observed one hour following consumption of a single serving of 50 g freeze-dried strawberry powder, providing 142 mg ACN and given as a strawberry drink, in a group of 34 overweight or obese individuals [90]. However, when the same serving of freeze-dried strawberry powder was given to the same individuals daily for a period of 4 weeks, no effect on SBP or DBP was observed [90].

In a double-blind, placebo-controlled, parallel arms intervention, 26 patients with diabetes received two daily cups of a drink made with 50 g freeze-dried strawberry powder. After 6 weeks, a significant reduction in DBP, but not SBP, was observed [91].

In a pre-post design intervention, daily consumption of two cups strawberry drink (50 g freeze-dried powder) for 4 weeks had no effect on SBP or DBP in a group of 16 women with metabolic syndrome [92].

In a subsequent controlled, parallel arms trial, daily consumption of twice that dose (four cups strawberry drink made of 100 g freeze-dried powder) for twice the time (8 weeks), still had no effect on SBP or DBP in a group of 27 patients with metabolic syndrome [93].

In yet another controlled, parallel arms experiment on 60 participants with risk factors for CVD, daily consumption of either 25 g or 50 g freeze-dried strawberry powder for 12 weeks had no effect on SBP or DBP [94]. In a subsequent placebo-controlled, crossover intervention, daily consumption of either 13 or 32 mg freeze-dried strawberry powder (providing 38 or 92 mg ACN, respectively) for 4 weeks, did not affect SBP or DBP in a group of 33 participants with obesity and dyslipidemia [95].

In a double-blind, placebo-controlled, parallel arms intervention, 60 post-menopausal women with pre- or stage 1 hypertension, received either 25 or 50 mg of freeze-dried strawberry powder, providing 102 or 204 mg of ACN, daily for 8 weeks. No effect on DBP was observed, while the decrease in SBP only reached statistical significance in the group receiving the lower dose [96].

In a controlled, crossover trial, 4 weeks consumption of a daily serving of 454 g strawberries, did not affect SBP or DBP in a group of 28 participants with hyperlipidemia [97].

In a double-blind, placebo-controlled, crossover intervention, no effect on SBP or DBP was observed following 12 weeks consumption of a daily serving of 50 g freeze-dried strawberry powder, in 17 participants with knee osteoarthritis [98].

In a double-blind, controlled, crossover intervention, 7 weeks consumption of a daily serving of 320 g of whole frozen strawberries, did not affect SBP or DBP in a group of 20 obese participants [99].

In conclusion, only three out of twelve studies reported a BP lowering effect associated with strawberry consumption. Only one of these studies specifically investigated hypertensive subjects, reporting only a slightly positive effect on BP. Overall, the association between strawberry consumption and BP regulation in humans appears to be very weak.

#### 3.2.3. Chokeberry

In a double-blind, placebo-controlled, parallel arms intervention with 101 overweight participants, a capsule with 90 or 150 mg chokeberry extract (16 or 27 mg ACN) was given daily for 24 weeks, resulting in a reduction in DBP which was greater in the groups receiving the higher dose [100].

In a pre-post intervention, daily consumption of chokeberry extract providing 300 mg of ACN, for 8 weeks, resulted in a significant reduction of SBP and DBP in 25 patients with metabolic syndrome [101].

In a double-blind, placebo-controlled, parallel arms trial, no effect on SBP or DBP was observed in a group of 66 healthy young male participants, following daily consumption of polyphenol-rich chokeberry extract (30 mg ACN) or whole chokeberry powder (4 mg ACN) for 12 weeks [102]. However, a positive effect on vascular function and gut microbiota composition was observed with both treatments.

In a pre-post intervention, 4 weeks consumption of a daily 100 mL serving of glucomannan-enriched (2 g) chokeberry juice-based supplement (25 mg ACN), lowered SBP but not DBP in a group of 20 post-menopausal women with abdominal obesity [103]. In a subsequent study with the same experimental design, 12 weeks consumption of the same juice did not affect SBP or DBP in a group of 29 healthy women [104].

In a pre-post intervention with 200 mL of polyphenol-rich organic chokeberry juice (358 mg ACN) given daily for 4 weeks to a group of 23 participants with pre- or stage 1 hypertension, resulted in a significant reduction of SBP, DBP and average 24 h BP [105].

In a single-blind, placebo-controlled, crossover trial, 37 participants with mild hypertension received daily servings of cold-pressed chokeberry juice and oven-dried chokeberry powder, providing a total of 1024 mg ACN, for 16 weeks, resulting in a significant reduction of daytime DBP, but not SBP [106].

In a double-blind, placebo-controlled, parallel arms study, daily consumption of a 255 mg chokeberry flavonoid extract (64 mg ACN) for 6 weeks, resulted in a significant reduction of both SBP and DBP in a group of 44 myocardial infarction survivors receiving statin therapy [107].

In a double-blind, placebo-controlled, parallel arms trial, no effect on SBP or DBP was observed following daily consumption or 100 mL high-polyphenols (113 mg ACN) or 100 mL low-polyphenols (28 mg ACN) chokeberry juice for 4 weeks, in participants with CVD risk factors [108].

In a pre-post intervention, 23 patients with untreated metabolic syndrome received a daily chokeberry extract supplement providing 60 mg ACN for 8 weeks, resulting in a significant reduction of both SBP and DBP. Furthermore, a significant reduction in ACE activity was observed, although still higher compared to a reference group of healthy controls and to another reference group of metabolic syndrome controls receiving ACE-inhibitors’ therapy [109].

In a pre-post intervention, 58 male participants with mild hypercholesterolemia received a daily 250 mL serving of chokeberry juice providing 90 mg ACN for two 6-week periods, separated by a 6 week wash-out. A significant reduction in both SBP and DBP was observed at the end of the second intervention, with DBP being already significantly lower at the end of the first intervention period [110].

In a pre-post intervention, a daily serving of 300 mL chokeberry extract (120 mg ACN) given for 4 weeks to a group of 143 participants with metabolic syndrome, significantly decreased both SBP and DBP compared to baseline [111].

In a placebo-controlled, parallel arms trial with 49 healthy former smokers, daily consumption of 500 mg chokeberry extract (45 mg ACN) for 12 weeks did not affect SBP or DBP [112].

In conclusion, the effect of chokeberry consumption on BP was extensively tested, with a wide range of dose (16 mg ACN to 1024 mg) and time (4 to 24 weeks) interventions.

A significant BP lowering effect was reported in 9 out of 13 studies, with doses as low as 25 mg and interventions as short as 4 weeks.

Of the four studies not reporting an effect, three were conducted on healthy participants, and one on healthy participants with CVD risk factors. In contrast, all studies reporting an effect were conducted on subjects with either overweight/obesity, hypertension, hypercholesterolemia, or metabolic syndrome.

Two studies specifically tested hypertensive patients, both reporting a BP lowering effect of chokeberry consumption.

All considered, these results suggest a very promising potential for the effect tof chokeberry consumption on BP control.

#### 3.2.4. Sweet and Sour (Tart) Cherry

In a crossover design experiment, a single serving of sweet cherry juice providing 207 mg of ACN, resulted in a significant reduction of SBP and DBP two hours after consumption in a group of 6 young and 7 older adults. Interestingly, this effect was only observed when the cherry juice was given in a single dose, but not when the same dose was split in three smaller servings given one hour apart [113].

In a subsequent controlled, parallel arms trial, a daily serving of sweet cherry juice (138 mg ACN) given for 12 weeks to a group of 49 older adults, significantly decreased SBP but not DBP [114].

Conversely, in a pre-post experiment, a daily serving of 280 g fresh sweet cherries given for 4 weeks to a group of 18 healthy participants, did not affect SBP or DBP at the end of the intervention or one month after its end [115].

In a single-blind, placebo-controlled, crossover study, a single serving of sour cherry juice (73.5 mg ACN) was able to significantly lower SBP and MAP, but not DBP, in a group of 15 men with early hypertension [116]. In subsequent studies with the same experimental design, SBP was also lowered in a group of 27 healthy middle-aged participants [117], as well as in a group of 10 young athletes, with no effect on MAP or DBP [118].

Following six weeks’ consumption of a daily serving of sour cherry juice providing 720 mg of ACN, a group of 19 women with diabetes had significantly lower SBP and DBP values compared to baseline, although no control group was present [119].

In a controlled, parallel arms intervention with 34 overweight older adults receiving a daily serving of sour cherry juice (451 mg total phenolics) for 12 weeks, there was a significant reduction of SBP but not DBP [120].

In a single-blind, placebo-controlled, parallel arms trial, 11 healthy participants received a daily serving of tart cherry juice (540 mg ACN) for 20 days, but no effect on SBP or DBP was observed, before or after performing physical exercise [121].

In a subsequent single-blind, placebo-controlled, crossover intervention, a daily serving of sour cherry juice providing half the amount of ACN given to a group of 12 participants with metabolic syndrome for one week, did not affect SBP, DBP or MAP, but resulted in a significant reduction in 24-h ambulatory SBP, DBP and MAP [122].

In a single-blind, placebo-controlled, parallel arms intervention, 19 participants with metabolic syndrome received a twice daily serving of 140 mL sour cherry juice (176 mg ACN) for 12 weeks, but no effect was observed in either central or peripheral SBP, DBP or MAP [123].

In a double-blind, placebo-controlled, parallel arms intervention, a twice daily serving of sour cherry juice (74 mg ACN) for 4 weeks, did not affect SBP, DBP or MAP in a group of 23 healthy young participants [124]. In a subsequent study with the same experimental design, with the same serving of sour cherry juice given for 3 months to a group of 40 middle-aged, overweight participants, it also did not affect SBP, DBP or MAP [125].

In a control, parallel arms trial, no effect on SBP or DBP was observed when a daily serving of sour cherry concentrate (274.5 mg ACN) was given for 6 weeks to 47 healthy adults [126].

In conclusion, some BP lowering effect of cherry consumption was observed in all four single dose studies, and in four out of ten longer term studies. However, five out of six of the studies that did not report an effect were all conducted on healthy participants, whereas the four studies reporting an effect were conducted on participants with overweight, diabetes, metabolic syndrome, or higher baseline BP values due to older age.

Finally, dose and duration do not appear to make a difference: a BP lowering effect was observed with doses as low as 70 mg ACN, and as high as 720 mg ACN per serving, and after a single serving of cherries.

### 3.3. Family Grossulariaceae

To the family Grossulariaceae belong berries of the genus *Ribes*, including blackcurrants (*R. nigrum*), redcurrants (*R. rubrum*) and gooseberries (*R. uva-crispa*). Within this family, blackcurrant is the richest source of vitamin C and polyphenols (up to 1% in weight), and for this reason it has been more widely studied. It has an average ACN content (around 200 mg/100 g) and is also rich in flavonols, especially myricetin [12].

#### Blackcurrant

In a double-blind, crossover intervention, a single dose of blackcurrant extract drink providing either 150, 300 or 600 mg of ACN following a high-carbohydrate meal, did not observe any effect on SBP or DBP after two hours in a group of 23 healthy participants [127].

In a controlled, crossover intervention on 15 athletes receiving a daily dose of blackcurrant extract providing either 105, 210 or 315 mg of ACN for 7 days, a significant reduction of MAP, but not of SBP or DBP, was observed with 210 or 315 mg of ACN, but not with 105 mg or in controls, suggesting a dose-dependent effect [128].

When the experiment was repeated on 13 healthy young males, in a double-blind, placebo-controlled, crossover intervention, testing only the 315 mg dose administered daily for one week, no effect was observed on SBP, DBP or MAP at rest, but a significant reduction of SBP, DBP and MAP was observed during sustained isometric contraction [129], suggesting an ability of blackcurrant to prevent BP spikes.

In yet another study, the authors tested the effect of a daily dose of 600 mg blackcurrant extract providing 210 mg of ACN, administered for one week to a group of 14 older adults, in a double-blind, placebo-controlled, crossover intervention. Both SBP and DBP, which were high at baseline, significantly decreased [130].

In a placebo-controlled, parallel arms intervention, administration of a daily dose of blackcurrant juice providing either 40 or 143 mg of ACN for six weeks to a group of 66 healthy adults, did not result in any effect on SBP or DBP [131].

In a double-blind, placebo-controlled, crossover intervention, 11 healthy participants received for two weeks a daily blackcurrant extract providing 7.7 mg of ACN per kg of body weight, and no effect on SBP or DBP was observed after a 30 min typing workload [132].

In a double-blind, placebo-controlled, crossover intervention, two daily 300 mg capsules of blackcurrant extract providing 210 mg of ACN were administered for one week to a group of 14 older adults, resulting in a significant reduction of both central and brachial SBP, as well as brachial DBP and MBP [133].

Overall, the studies on blackcurrant suggest the existence of a dose-dependent effect on BP. No effect was observed in studies with low doses (providing less than 200 mg of ACN), while a consistent effect was observed in studies with higher doses (providing more than 200 mg of ACN). Furthermore, an effect was only observed in older subjects with higher BP baseline values, or in healthy subjects when performing BP-raising physical efforts.

### 3.4. Family Arecaceae

To the family Arecaceae belong açai berries (*Euterpe oleracea*), abundant sources of ACN and condensed tannins [138].

#### Açai Berry

Three human interventions have measured the effect of açai berry consumption on BP.

A single dose of açai-based smoothie providing 493 mg of ACN given to 23 healthy males following a high-fat meal, improved vascular function but did not have any effect on SBP or DBP after two or six hours [134].

A two-month, double-blind, placebo-controlled intervention with 200 g of açai pulp added daily to the hypo energy-containing diet of 69 overweight, dyslipidemic participants, had a positive outcome on inflammation but did not affect SBP or DBP [135].

A small, pre-post trial with 100 g of açai pulp given daily for four weeks to a group of 10 overweight participants, also failed to observe any effect on SBP or DBP [136].

In conclusion, none of the available human interventions investigating the effect of açai berries in humans reported an effect on BP.

## 4. Discussion

Evidence for an effect of berries on BP is presented in the case of chokeberries, which are characterized by the highest total phenolic content, one of the highest ACN contents, high condensed tannins content and very good flavonols content, especially quercetin.

Encouraging, albeit so far inconsistent, evidence is also available for cherries, blackcurrants, blueberries and raspberries.

Given the relatively large number of studies available, the two berries reporting the most inconclusive evidence for an effect on BP are cranberries and strawberries. Interestingly, these are the two berries with lowest ACN content, suggesting that the presence of ACN may be the key factor when it comes to the relation between berry consumption and affecting BP.

Other berries for which clinical studies show no effect on BP are bilberries and açai berries, although in this case only a limited number of studies are available (three studies each), and none of them specifically targeted BP or hypertensive subjects.

It should be noted, however, that all of the studies in this review either reported a blood-pressure lowering effect, or no effect, while no study reported a hypertensive effect associated with berry consumption. This suggests that berry consumption has either no effect or a positive effect on BP, but never a detrimental effect.

With few exceptions, the study design did not involve a washout period before the start of the intervention, during which participants would be asked to follow a low-polyphenols diet, or to abstain from berry consumption. This means that berry treatment was in addition to the normal diet of the participants thus strengthening the ecological validity of those studies that report changes in blood pressure.

The effects of dose and duration appear to be associated with the type of berry and may play a lesser role. Positive effects were reported with a wide range of doses with chokeberries and cherries. Some evidence for a dose effect was found in studies with raspberries and blueberries, and even more consistently in studies with blackcurrants. However, while some studies suggest the existence of a dose effect (a minimum dose is required to exert an effect), no evidence for dose-dependency (larger effect with increasing dosages) is available. As far as duration is concerned, studies suggest a significant effect in acute, single-dose studies as well as in longer term/chronic ones.

Rather than dose, duration or type of berry, a more significant source of variability in the observed results is likely to derive from two major limitations of the studies that are available.

First, BP varies during the day, and as reviewed above, most studies only measure it at one time point (usually in the morning). Indeed, in the very small number of studies that measured 24-h ambulatory BP, there was no significant reduction in a single-point, morning BP measurement, but there was a significant reduction in the average 24-h measurement. This suggests the possibility that in many of the studies that did not observe an effect on SBP or DBP, a more significant effect may have been found if they had measured BP for 24-h.

Second, in most of the studies reviewed in this article, BP was not among the primary outcomes of the intervention. Since it is an easy and inexpensive measurement, many studies nevertheless often report it as a general indicator of the participants’ status. While this increases the number of clinical trials available for this marker, it decreases the significance of the measure in that participants are usually normotensive.

Indeed, BP was the primary outcome only in 8 of the reviewed interventions (one with blueberries, cranberries, raspberries, strawberries, cherries and berry mix and two with chokeberries), and all of these studies reported at least some positive effect on BP.

This is likely because these trials were conducted in subjects that were pre-hypertensive or hypertensive at recruitment. Indeed, a recurring observation is that the effect on BP is more evident when baseline values are higher; for example, when only the subset of hypertensive or older participants is considered, or when participants are asked to smoke a cigarette or to perform a BP-raising physical activity.

Another recurring observation is that in studies conducted on healthy subjects, there is no effect on BP. The situation is less clear when participants with overweight/obesity, metabolic syndrome, or diabetes are involved. In many cases, studies with these patients failed to find an effect on BP.

The fact that in studies reporting a decrease in BP participants had high baseline values, while no effect was ever found in participants that were normotensive at recruitment, suggests that berries indirectly act to lower BP in hypertensive subjects by improving vascular function. This seems confirmed by the observation that several studies not reporting any direct effect on BP, reported positive effects on markers of endothelial function (such as flow mediated dilation or NO levels) that are known to indirectly lower BP.

From a clinical point of view, this is an encouraging observation, suggesting that any potential hypotensive effect is not exerted regardless and indiscriminately, but only when it is actually advantageous (that is, only in hypertensive subjects). Furthermore, an improvement of vascular function, independent of whether it results in lowered BP, is nevertheless beneficial in terms of cardiovascular health, through improved blood-flow, endothelial reactivity, reduced inflammation, and secretion of mediators related to vascular homeostasis and remodeling.

Finally, it needs to be pointed out that the metabolic fate of ACN during digestion and following absorption is intricate and still largely unknown. The vast majority are not absorbed intact, but go through hydrolysis and partial degradation to other phenolic compounds or following fermentation by the gut microbiota. Once in the bloodstream, whole ACN, their parent compounds, degradation products and microbial metabolites are all quickly converted to methyl, glucuronide and sulfate conjugated metabolites by phase I and phase II enzymes.

Thus, it is likely that most of the biological responses to ACN consumption are due to the activity of secondary phase metabolites, and that differences in ACN metabolism account for a large proportion of the observed variability in blood pressure results. This may involve not only individual differences in ACN metabolism, but also different metabolic responses related to different doses, frequency of intake, and time duration of interventions. This is a factor that will need to be kept in mind and more carefully investigated in future studies.

## 5. Conclusions

In conclusion, while berry consumption has been more consistently associated with positive effects on inflammation or blood lipid regulation in clinical studies, its effect on BP is less consistent.

When such an effect is observed, it is only in subjects with high baseline values of BP, suggesting that berries do not have a direct hypotensive effect, but a potential for BP normalization in hypertensive subjects via the modulation of endothelial and vascular function.

There is also weak evidence suggesting a specific role of berries in the prevention of hypertension, other than the effect associated with fruit and vegetable consumption in general.

Among berries, the review of clinical trials suggests a stronger effect on BP for chokeberries, encouraging but mixed evidence for cherries, blackcurrants, blueberries and raspberries, weaker evidence for cranberries and strawberries, and not enough evidence to draw conclusions for bilberries and açai berries.

At the present state of knowledge, while regular consumption of berries is indeed to be encouraged for their multiple health benefits, BP regulation does not appear to be the main consideration for promoting their consumption.

Further investigations specifically targeted to hypertensive subjects are warranted to pinpoint the mechanisms involved and to better assess the extent of the BP lowering effect associated with berry consumption, as well as the most appropriate dosages and type of berries to be recommended.

## Figures and Tables

**Figure 1 nutrients-14-02701-f001:**
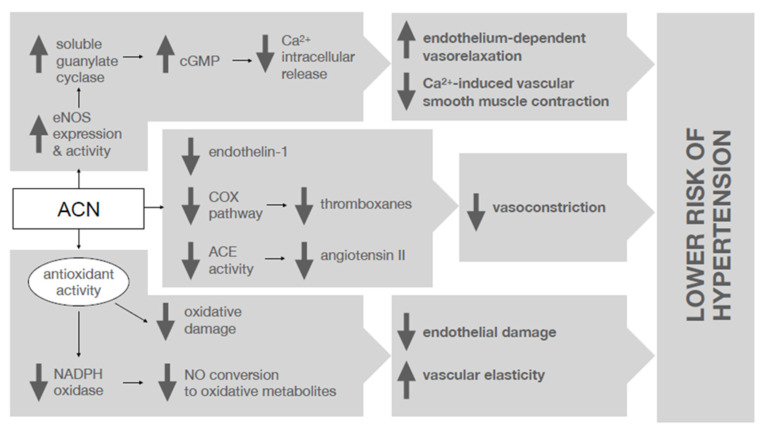
Potential mechanisms by which anthocyanins may improve vascular function and prevent endothelial damage. ACE, angiotensin converting enzyme; ACN, anthocyanins; cGMP, Cyclic guanosine monophosphate; COX, cyclooxygenase; eNOS, endothelial nitric oxide synthase; NADPH, nicotinamide adenine dinucleotide phosphate; NO, nitric oxide.

**Table 1 nutrients-14-02701-t001:** Summary of findings related to blood pressure in berry intervention trials.

Berry	Treatment Duration (*)	Study Design (**)	Number of Participants	Characteristics of Participants (***)	Treatment (****)	Effect on Blood Pressure (*****)	References
Blueberry (highbush)	SD	C, cross	10	Healthy, age 21 ± 2, BMI 23 ± 2	Blueberry drink from FDP (348 mg ACN)	=SBP, =DBP	[56]
Blueberry (highbush)	SD	C, par	16	Smokers, age 24 ± 1, BMI 23 ± 1	Blueberry drink from FDP (348 mg ACN), followed by smoking 1 cigarette	↓SBP post smoke, = DBP	[56]
Blueberry (highbush)	SD	C, cross	12	M, healthy, age 24 ± 1, BMI 23 ± 1	Drink from FDP (309 mg ACN)	=SBP, =DBP	[57]
Blueberry (highbush)	SD	C, cross	12	M, smokers, age 24 ± 1, BMI 23 ± 1	Drink from FDP (309 mg ACN) followed by smoking 1 cigarette	=SBP, =DBP	[57]
Blueberry (highbush)	SD	DB, C, cross	10	M, healthy, age 27 ± 3, BMI 25 ± 3	Drink from FDP (310 mg ACN)	=SBP, =DBP	[58]
Blueberry (highbush)	SD	DB, C, cross	10	Healthy, age 27 ± 1, BMI 25 ± 1	Drink from FDP (330 mg ACN), or baked product with same amount of FDP (196 mg ACN)	=SBP, =DBP	[59]
Blueberry (highbush)	8 W	DB, PC, par	48	W, postmenopausal, with pre- and stage-1 hypertension, age 59 ± 5, BMI 31 ± 6	480 mL drink from 22 g FDP (469 mg ACN)	↓SBP, ↓DBP	[60]
Blueberry (highbush)	8 W	SB, C, par	48	With MetS, age 50 ± 3, BMI 38 ± 2	480 mL drink from 50 g FDP (742 mg ACN)	↓SBP, ↓DBP	[61]
Blueberry (highbush)	6 W	PC, par	25	Healthy, age 43 ± 12, BMI 26 ± 4	FDP [eq. 250 g fresh berries]	↓aortic systolic pressures (ASPs), ↓SBP, =DBP, ↓DBP in subset of 9 pre-hypertensive subjects	[62]
Blueberry (highbush)	18 W	C, par	34	W, in early pregnancy with history of gestational diabetes, age 27 ± 5, BMI 36 ± 4	280 g whole frozen blueberries (700 mg ACN) + 12 g soluble fiber supplement per day	=SBP, ↓DBP	[63]
Blueberry (highbush)	4 M	DB, PC, par	63	With knee osteoarthritis, age 56 ± 1, BMI 32 ± 1	40 g FD whole blueberry powder daily	↓SBP, =DBP	[64]
Blueberry (wild)	6 M	DB, PC, par	122	Older adults, age 71 ± 4, weight 71 ± 4 kg	1 or 2 g FDP, or 200 mg extract (2.7, 5.4 or 14 mg ACN)	↓SBP with extract, but not with powders, at 3 and 6 M	[65]
Blueberry (highbush)	6 M	DB, PC, par	115	With overweight/obesity and MetS, age 63 ± 7, BMI 31 ± 3	13 or 26 mg/day FDP (182 or 364 mg ACN)	=SBP, =DBP	[66]
Blueberry (highbush)	3 W	C, par	20	Smokers, age 28 ± 4, BMI 29 ± 3	250 g fresh berries	=SBP, =DBP	[67]
Blueberry (highbush)	6 W	DB, PC, par	32	With obesity and insulin resistance, age 52 ± 3, BMI 37 ± 1	45 g FDP added to smoothie and yogurt (580 mg ACN)	=SBP, =DBP	[68]
Blueberry (highbush)	6 W	DB, PC, par	44	With MetS, age 57 ± 2, BMI 36 ± 1	45 g FDP added to smoothie and yogurt (580 mg ACN)	=SBP, =DBP	[69]
Blueberry (wild)	6 W	PC, cross	18	M, with risk factors for CVD, age 48 ± 10, BMI 25 ± 3	250 mL drink from 25 g FDP (400 mg ACN)	=SBP, =DBP	[70]
Blueberry (wild)	1 W	SB, PC, cross	19	W, with T2DM risk, age 39–64, BMI 27–37	240 mL juice	=SBP, =DBP	[71]
Bilberry	4 W	DB, PC, cross	20	With T2DM, age 52. ± 3, BMI 27 ± 2	1.4 g/day extract	=SBP, =DBP	[72]
Bilberry	8 W	C, par	27	With MetS, age 51 ± 6, BMI 32 ± 4	200 g puree + 40 g dried (1381 mg ACN)	=SBP, =DBP	[73]
Bilberry	6 W	Pre-post	36	Healthy, age 48 ± 6, BMI 27 ± 4	Frozen berries, 3 times a week (456 mg ACN)	=SBP, =DBP	[74]
Cranberry	SD	C, cross	40	With obesity and T2DM, age 56 ± 6, BMI 40 ± 7	40 g FDP following high fat meal, test after 1 h, 2 h and 4 h	=SBP, =DBP	[75]
Cranberry	SD	Pre-post	15	With overweight/obesity and coronary artery disease, age 62 ± 8, BMI ?	480 mL juice following meal	=SBP, =DBP	[76]
Cranberry	4 W	PC, cross	44	With overweight/obesity and coronary artery disease, age 62 ± 10, BMI 29 ± 4	480 mL/day juice	=SBP, =DBP	[76]
Cranberry	8 W	DB, PC, par	31	W, with MetS, age 52 ± 8, BMI 40 ± 7	480 mL/day juice	=SBP, =DBP	[77]
Cranberry	8 W	DB, PC, par	53	With obesity and elevated fasting glucose or impaired glucose tolerance, age 48 ± 14, BMI 37 ± 5	450 mL/day low-energy berry beverage (6.5 mg ACN)	=SBP, =DBP	[78]
Cranberry	12 W	DB, PC, par	30	With T2DM, age 65.5 ± 2, BMI 26 ± 1	500 mg/capsule FDP	=SBP, =DBP	[79]
Cranberry	8 W	PC, par	56	With overweight/obesity, age 50 ± 11, BMI 28 ± 4	Twice daily juice (173 or 62 mg TP)	=SBP, ↓DBP	[80]
Cranberry	8 W	PC, cross	40	With overweight/obesity and pre-hypertension, age 47 ± 12, BMI 29 ± 5	500 mL/day drink (27% cranberry juice)	=SBP, =DBP, but ↓ 24-h ambulatory DBP during daytime hours.	[81]
Cranberry	4 W	DB, PC, cross	35	M, with abdominal obesity, with (13) or without MetS, age 45 ± 10, BMI 28 ± 2	500 mL/day of low-calorie drink (27% juice)	=SBP, =DBP	[82]
Cranberry	2 W	Pre-post	21	M, with dyslipidemia and abdominal obesity, BMI 27 ± 4, age 38 ± 8	7 mL/kg BW (range 460–760 mL/day berry juice)	=SBP, =DBP	[83]
Cranberry	12 W	Pre-post	30	M, with abdominal obesity, with (9) or without MetS, age 51 ± 10, BMI 28± 3	125 mL/day juice (4 W) + 250 mL/day (4 W) + 500 mL/day (4 W)	↓SBP with highest dose, =DBP	[84]
Raspberry (red)	4 W	C, cross	22	With obesity, age 54 ± 4, BMI 33 ± 2	Frozen berries (225 mg ACN)	↓SBP, =DBP	[85]
Raspberry (red)	SD	C, cross	25	With obesity and T2DM, age 54 ± 4, BMI 35 ± 2	Post-prandial assessment with or without frozen berries (225 mg ACN)	=SBP, =DBP	[85]
Raspberry (red)	8 W	C, par	50	With overweight/abdominal obesity and slight hyperinsulinemia or hypertriglyceridemia, BMI 30 ± 5, age 32 ± 9	280 g/day of frozen berries	=SBP, =DBP	[86]
Raspberry (black)	8 W	DB, PC, par	45	With pre-hypertension, age 57 ± 12, BMI 25 ± 3	1500 mg or 2500 mg daily FDP	↓SBP with high dose, =DBP	[87]
Raspberry (black)	12 W	DB, PC, par	51	With MetS, age 59 ± 10, BMI 25 ± 4	750 mg daily FDP	=SBP, =DBP	[88]
Strawberry	SD	PC, cross	30	With overweight/obesity, age 28 ± 2, BMI 31 ± 1	40 g FDP + high-fat meal [eq. 450 g fresh berries]	=SBP	[89]
Strawberry	4 W	DB, C, par	34	With overweight/obesity, age 53 ± 5, BMI 31 ± 5	Twice daily drink from 25 g FDP each (total 142 mg ACN)	=SBP, =DBP	[90]
Strawberry	SD	DB, C, par	34	With overweight/obesity, age 53 ± 5, BMI 31 ± 5	Drink from 50 g FDP (142 mg ACN total)	↓SBP 1 h post treatment	[90]
Strawberry	6 W	DB, PC, par	36	With T2DM, BMI 28± 4, age 51 ± 11	2 cups/day drink from 25 g FDP each (142 mg ACN total)	=SBP, ↓DBP	[91]
Strawberry	4 W	Pre-post	16	W, with obesity and MetS, age 39–71, BMI 39 ± 2	2 cups/day drink from 25 g FDP each (142 mg ACN total)	=SBP, =DBP	[92]
Strawberry	8 W	C, par	27	With obesity and MetS, age 47 ± 3, BMI 37 ± 2	4 cups/day drink from 25 g FDP each (284 mg ACN total)	=SBP, =DBP	[93]
Strawberry	12 W	C, par	60	With CVD risk factors, age 49 ± 10; BMI 36 ± 5	2 cups/day drink with low dose (25 g) or high dose (50 g) FDP (142 or 284 mg ACN)	=SBP, =DBP	[94]
Strawberry	4 W	PC, cross	33	With obesity and dyslipidemia, age 53 ± 13, BMI 33 ± 3	13 or 32 mg/day FDP (38 or 92 mg ACN)	=SBP, =DBP	[95]
Strawberry	8 W	DB, PC, par	60	W, postmenopausal, with pre- or stage 1 hypertension, age 59 ± 8, BMI 32 ± 7	25 or 50 mg/day FDP (102 mg or 204 mg ACN)	↓SBP with 25 mg, =DBP	[96]
Strawberry	4 W	C, cross	28	With dyslipidemia, age 38–75, BMI 20–32	454 g/day fresh strawberries	=SBP, =DBP	[97]
Strawberry	12 W	DB, PC, cross	17	With knee osteoarthritis, age 57 ± 7, BMI 39 ± 2	Twice daily drink from 50 g FDP [eq. 500 g fresh berries]	=SBP, =DBP	[98]
Strawberry	7 W	DB, C, cross	20	With obesity, age 20–50, BMI 30–40	Two servings of FDP mixed as a milkshake, in yogurt, cream cheese, or water [eq. 320 g frozen berries]	=SBP, =DBP	[99]
Chokeberry	24 W	DB, PC, par	101	With overweight/obesity, age 53 ± 10, BMI 29 ± 5	90 mg or 150 mg berry extract capsules (16 mg or 27 mg ACN)	↓DBP with 150 compared to 90 mg	[100]
Chokeberry	8 W	Pre-post	25	With MetS, age 42–65, BMI 31 ± 3	Berry extract (300 mg ACN)	↓SBP, ↓DBP	[101]
Chokeberry	12 W	DB, PC, par	66	M, healthy, age 24 ± 5, BMI 23 ± 2	Capsules of polyphenol-rich extract (30 mg ACN) or whole chokeberry powder (4 mg ACN)	=SBP, =DBP	[102]
Chokeberry	4 W	Pre-post	20	W, postmenopausal, with abdominal obesity, age 53 ± 5, BMI 36 ± 4	100 mL/day glucomannan-enriched (2 g), berry juice (25 mg ACN)	↓SBP, =DBP	[103]
Chokeberry	12 W	Pre-post	29	W, healthy, age 35 ± 8, BMI 23 ± 4	100 mL/day glucomannan-enriched (2 g), berry juice (25 mg ACN)	=SBP, =DBP	[104]
Chokeberry	4 W	Pre-post	23	With pre- or stage-1 hypertension, age 48 ± 10, weight 82 ± 20	200 mL/day of polyphenol-rich organic berry juice (358 mg ACN)	↓SBP, ↓DBP, ↓ average 24 h BP	[105]
Chokeberry	16 W	SB, PC, cross	37	With mild hypertension, age 40–70, BMI 26 ± 3	Cold-pressed berry juice and oven-dried berry powder (1024 mg ACN total)	↓daytime DBP (recorded over 15 h), =SBP	[106]
Chokeberry	6 W	DB, PC, par	44	Post myocardial infarction patients, receiving statin therapy, age 66 ± 8, BMI 27 ± 3	255 mg/day berry polyphenol-rich extract (64 mg ACN)	↓SBP, ↓DBP	[107]
Chokeberry	4 W	DB, PC, par	84	With CVD risk factors, age 41 ± 8, BMI 27 ± 6	100 mL/day high-polyphenols or 100 mL/day low-polyphenols berry juice (113 mg or 28 mg ACN)	=SBP, =DBP	[108]
Chokeberry	8 W	Pre-post	23	23 with untreated MetS (BMI 31 ± 4), reference group with 25 treated MetS patients (BMI 29 ± 3) and 20 healthy controls (BMI 23 ± 1)	Berry extract (60 mg ACN), or ACE-inhibitors	↓SBP, ↓DBP	[109]
Chokeberry	6 W + 6 W	Pre-post	58	M, with mild hypercholesterolemia, age 54 ± 6, BMI 28 ± 3	250 mL/day berry juice (6-week intervention + 6-week wash-out + 6-week intervention) (90 mg ACN)	↓SBP after 12 W, ↓DBP after 6 and 12 W	[110]
Chokeberry	4 W	Pre-post	143	With MetS, BMI 32 ± 6, age 55 ± 15	30 mL/day berry extract (120 mg ACN)	↓SBP, ↓DBP	[111]
Chokeberry	12 W	PC, par	49	Healthy former smokers, age 35 ± 3, BMI 26 ± 1	500 mg berry extract (45 mg ACN)	=SBP, =DBP	[112]
Cherry (sweet)	SD	Cross	13	6 young (age 22 ± 1, BMI 26 ± 4) and 7 older adults (age 78 ± 6, BMI 29 ± 4)	Juice (207 mg ACN), in single dose or split into three doses over 2 h	↓SBP, ↓DBP at 2 h after consumption, if given in a single dose (but not if split in three doses given 1 h apart)	[113]
Cherry (sweet)	12 W	C, par	49	Older adults, age 80 ± 6, BMI 26 ± 3	Juice (138 mg)	↓SBP, =DBP	[114]
Cherry (sweet)	4 W	Pre-post	18	Healthy, age 50 ± 4, BMI 26 ± 4	280 g fresh fruit	=SBP, =DBP at the end of the trial and after 1 month	[115]
Cherry (tart)	SD	SB, PC, cross	15	M, with early hypertension, age 31 ± 9, BMI 27 ± 4	Juice (74 mg ACN)	↓SBP, ↓MAP, = DBP	[116]
Cherry (tart)	SD	DB, PC, cross	27	Healthy, age 50 ± 6, BMI 26 ± 5	Concentrate (68 mg ACN)	↓SBP	[117]
Cherry (tart)	SD	DB, PC, cross	10	Athletes, age 28 ± 7, weight 78 ± 9 kg	Juice (68 mg ACN)	↓SBP, =DBP, =MAP	[118]
Cherry (tart)	6 W	Pre-post	19	W, with T2DM, age 53 ± 9, BMI 30 ± 4	Juice (720 mg ACN)	↓SBP, ↓DBP	[119]
Cherry (tart)	12 W	C, par	34	With overweight/obesity, age 70 ± 4, BMI 28 ± 4	Juice [451 mg TP]	↓SBP, =DBP	[120]
Cherry (tart)	3 W	SB, PC, par	11	Healthy, age 30 ± 10, BMI 24 ± 3	Juice (540 mg ACN)	=SBP, =DBP pre or post exercise	[121]
Cherry (tart)	1 W	SB, PC, cross	12	With MetS, age 50 ± 10, BMI 31 ± 7	Juice (270 mg ACN)	=SBP, =DBP, =MAP but ↓24-h ambulatory SBP, DBP and MAP.	[122]
Cherry (tart)	12 W	SB, PC, par	19	With MetS, age 36 ± 11, BMI 34 ± 8	240 mL juice, twice daily (176 mg ACN)	=SBP, =DBP, =MAP (both central and peripheral)	[123]
Cherry (tart)	4 W	DB, PC, par	23	Healthy, age 23 ± 3, BMI 25 ± 3	Twice daily 30 mL juice (74 mg ACN)	=SBP, =DBP, =MAP	[124]
Cherry (tart)	3 M	PC, par	50	With overweight/obesity, age 48 ± 6, BMI 28 ± 4	Twice daily 30 mL juice (74 mg ACN)	=SBP, =DBP, =MAP	[125]
Cherry (tart)	6 W	C, par	47	Healthy, age 38 ± 6, BMI 24 ± 3	Concentrate (275 mg ACN)	=SBP, =DBP	[126]
Blackcurrant	SD	DB, cross	23	Healthy, age 46 ± 14, BMI 26 ± 3.8	Drink from extract at different doses (150, 300 or 600 mg ACN), following high-carbohydrate meal	=SBP, =DBP after 2 h	[127]
Blackcurrant	1 W	C, cross	15	Athletes, age 38 ± 12, weight 76 ± 10 kg	Extract at different doses (105, 210 or 315 mg ACN)	= SBP, =DBP, ↓MAP with 210 and 315 mg	[128]
Blackcurrant	1 W	DB, PC, cross	13	M, healthy, age 26 ± 4, BMI 25 ± 3	Extract (315 mg ACN)	= SBP, =DBP, =MAP at rest; ↓SBP, ↓DBP, ↓MAP during isomeric contraction	[129]
Blackcurrant	1 W	DB, PC, cross	14	Older adults, age 69 ± 4, weight 85 ± 12 kg	600 mg/day extract (210 mg ACN)	↓SBP, ↓DBP	[130]
Blackcurrant	6 W	PC, par	66	Healthy or overweight, age 52 ± 10, BMI 29 ± 6	Juice, low or high dose (40 mg or 143 mg ACN)	=SBP, =DBP	[131]
Blackcurrant	2 W	DB, PC, cross	11	Healthy, age 39 ± 12, BMI ?	ACN-rich extract (7.7 mg ACN/kg of body weight)	=SBP, =DBP after 30 min typing workload	[132]
Blackcurrant	1 W	DB, PC, cross	14	Older adults, age 73 ± 6, BMI 22 ± 3	Two 300 mg capsules extract/day (35% blackcurrant extract) (210 mg ACN)	↓central SBP, ↓brachial SBP, DBP and MAP	[133]
Açai	SD	DB, C, cross	23	M, healthy, age 46 ± 9, BMI 28 ± 2	Berry smoothie following high-fat meal (493 mg ACN)	=SBP, =DBP at 2 and 6 h	[134]
Açai	2 M	DB PC, par	69	With overweight and dyslipidemia, age 41 ± 10, BMI 35 ± 6	200 g pulp with hypoenergetic diet [684 mg TP]	=SBP, =DBP	[135]
Açai	4 W	Pre-post	10	With overweight, age 28±?, BMI 27 ± 2	100 g pulp [0.77 mg/mL ACN in the pulp, density unknown]	=SBP, =DBP	[136]

(*) M, months; SD, single dose; W, weeks. (**) C, controlled without placebo; cross, crossover design; DB, double blind; par, parallel arms design; PC, placebo controlled; SB, single blind. (***) Age (in years) and BMI (in kg/m^2^) data expressed as mean ± SD, when available. A question mark (?) indicates unreported data. BMI, body mass index; CVD, cardiovascular disease; M, only male participants; MetS, metabolic syndrome; T2DM, type-II diabetes mellitus; W, only female participants. (****) ACN content is reported when available. If unreported, TP content is reported instead, if available. ACN, anthocyanins; FDP, freeze-dried powder; TP, total phenolics. (*****) Only blood pressure data are reported in this table: other outcomes of the studies are not reported. DBP, diastolic blood pressure; MAP, mean arterial pressure; SBP, systolic blood pressure.

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
