# Peer review of "The Role of Berry Consumption on Blood Pressure Regulation and Hypertension: An Overview of the Clinical Evidence"

_nutrients, 2022, doi:10.3390/nu14132701_

Round 1

Reviewer 1 Report

This paper titled “The role of berry consumption on blood pressure regulation and hypertension: an overview of the clinical evidence” is interesting.

My comments are as follow:

  1. The references are old, authors should focus on the articles published recent years.  
  2. Author should organize the data in tables and figures.
  3. English should be improved.

Author Response

We thank the reviewer for his/her comments and feedback.

1) The references are old, authors should focus on the articles published recent years

As stated in the manuscript, we have limited the review of clinical interventions to studies published in the last 20 years. We feel this is a reasonable timespan and that these publications are recent enough to be of interest. Older studies are only referenced when discussing potential mechanisms of action (mainly the study by Bhargava & Westfall on the mechanism of blood pressure depression by ellagic acid), when we think they are still relevant.

2) Author should organize the data in tables and figures.

We have now added a table with a summary of results and a figure depicting the potential mechanisms of actions of ACN.

3) English should be improved.

English has been checked thoroughly. In case there are further issues, it would be helpful if the reviewer could point out to some specific examples of mistakes or language style that (s)he deems unsatisfactory. 

Reviewer 2 Report

The review article (#nutrients-1787424) by Stefano Vendrame et al. took an effort in discussing the role of the chosen groups of polyphenols as a products of berries in the decrease of hypertension, and the maintenance of proper BP. The article is well-written and gives a clear picture of the actual state of knowledge but is difficult to follow. There are neither tables nor figures. It would be better to follow and read easily if Authors add any table or figure.

1.       Abstract. The term polyphenol is not well-defined, but is generally agreed that they are natural products  including four principal classes: phenolic acids, flavonoids, stilbenes, and lignans. Thus, it should be well mentioned.

2.       Authors have to introduce the part relating to the general presentation of polyphenols, where the proper division will be described, and then underlain that only anthocyanins, condensed tannins and ellagic acid are presented.

3.       When the abbr. is introduced for the first time, it should be explained: BP.

4.       Not whose excess but which

5.       “detrimental to blood pressure due to their unfavorable” – use abbr.

6.       Use BP along the whole text, there are many times when abbr. is not used

7.       These pooled data on over 155,000 men do relate to the ref1 7?

8.       P2 - DBP – as diastolic blood pressure is not explained

9.       ox LDL – explain it

10.   P 4 – SOD, GLUT, CAT explain please/ all GLUTs?

11.   P4 – please explain: The treatment inhibited IL-1b, NLRP3 and Caps-1 pathways

12.   P13 - BP depends on many factors not only changes during the day

13.   Please add tables and figure showing the role of the chosen polyphenols in BP regulation

Author Response

We thank the reviewer for his/her comments and feedback.

1.       Abstract. The term polyphenol is not well-defined, but is generally agreed that they are natural products  including four principal classes: phenolic acids, flavonoids, stilbenes, and lignans. Thus, it should be well mentioned.

We have now added the definition of the term to the text (beginning of section 2), as suggested. However, we feel it is not appropriate to add the definition of the term ‘polyphenols’ to the abstract, also considering that the abstract specifically mentions the specific classes of compounds that will be discussed in the article (anthocyanins, condensed tannins and ellagic acid).

2.       Authors have to introduce the part relating to the general presentation of polyphenols, where the proper division will be described, and then underlain that only anthocyanins, condensed tannins and ellagic acid are presented.

We have now rearranged the sentences according to this suggestion. 

3.       When the abbr. is introduced for the first time, it should be explained: BP.

4.       Not whose excess but which

5.       “detrimental to blood pressure due to their unfavorable” – use abbr.

6.       Use BP along the whole text, there are many times when abbr. is not used

8.       P2 - DBP – as diastolic blood pressure is not explained

9.       ox LDL – explain it

10.   P 4 – SOD, GLUT, CAT explain please/ all GLUTs?

11.   P4 – please explain: The treatment inhibited IL-1b, NLRP3 and Caps-1 pathways

Comments 3 through 6 and 8 through 11: we have modified the manuscript accordingly and homogenized abbreviations throughout.

7.       These pooled data on over 155,000 men do relate to the ref1 7?

Yes, we have now repeated the reference at the end of the first sentence to make this clearer.

12.   P13 - BP depends on many factors not only changes during the day

We re-checked the manuscript carefully, but we believe it is never said or implied that circadian variations are the only factors affecting blood pressure. It is just discussed as a potentially contributing factor.

13.   Please add tables and figure showing the role of the chosen polyphenols in BP regulation

We have now added a table with a summary of results and a figure depicting the potential mechanisms of actions of ACN.

Reviewer 3 Report

I have reviewed the manuscript titled “The role of berry consumption on blood pressure regulation and hypertension: an overview of the clinical evidence”. The manuscript identifies a topic of interest and combines findings from multiple different berries on blood pressure responses. The manuscript will be of interest to those who undertake cardiovascular research with berries and also nutritionists working with patients with hypertension. I provide some comments that the authors should address to improve their manuscript.

In places the manuscript is descriptive and lists findings of studies, without any critical appraisal of these. Furthermore, the manuscript often states that SBP or DBP were altered but gives no indication as to the magnitude of this change. Therefore, it is difficult for the reader to appreciate the magnitude of change occurring, or if the changes in SBP and DBP are clinically meaningful. It would help with clarity and appraisal if the authors are able to include this information.

It might help with clarity for the reader if the results were displayed in a table so that comparisons can be easily made.

I feel the authors need to be clear on the mode of delivery of the different berries given. For example, in a lot of the studies cited about blackcurrant, these were done with encapsulated spray-dried blackcurrant which allows for an easy consumption of a high anthocyanin intake. The way the current manuscript is written, it communicates as if participants in these studies consumed the berries. Consuming over 200 mg of anthocyanins from fresh blackcurrant would involve a large amount of blackcurrant which might be difficult for some participants to consume. Therefore, encapsulation makes it easier for participants to consume this amount of anthocyanins and also allows for blinding with placebo capsules.

Did any of the studies cited that examined blood pressure responses use a low polyphenolic diet before the laboratory visit (i.e. a washout diet before visiting the laboratory/ clinic)? A strength to many of the studies that didn’t use a washout diet, is that blood pressure changes are observed following supplementation of the berries on top of the participants normal diet – in turn, increasing ecological validity.

A key consideration that the authors should also refer to is the role of anthocyanin metabolites. Due to the bioavailability of whole anthocyanins being low, it is likely that the metabolites and secondary phase metabolites are causing the blood pressure responses. Therefore, this content could be included within the discussion as it is an important consideration. Especially for the studies demonstrating changes following different doses and time-durations.

Author Response

We thank the reviewer for his/her comments and feedback.

1) In places the manuscript is descriptive and lists findings of studies, without any critical appraisal of these. Furthermore, the manuscript often states that SBP or DBP were altered but gives no indication as to the magnitude of this change. Therefore, it is difficult for the reader to appreciate the magnitude of change occurring, or if the changes in SBP and DBP are clinically meaningful. It would help with clarity and appraisal if the authors are able to include this information.

Unfortunately, it is impossible to report such data in a uniform way that would make comparisons easy: it would actually clutter the manuscript. This is because some studies only report graphs (values would need to be guessed from the y axis), some report before and after values for both test and control groups (4 data points), some studies only the final values of test compared to controls, some studies only a before and after with no controls, some studies the percent variation, yet some other studies have multiple time points, some other test multiple dosages, and for some studies the relevant findings are only in a subgroup of subjects. Some of them evaluate blood pressure restoration following a challenge (eg cigarette smoking, or exercise), thus further increasing data points (before challenge, after challenge, after treatment).

Thus, we believe that reporting BP findings in a yes-or-no manner (statistically significant variation, or no statistically significant variation) is enough for the purpose of this review, while the reader who is interested in the exact terms and magnitudes of the variations for some specific studies, should refer to the original manuscripts. This actually prevents the risk of misinterpreting the significance of results, since it is difficult to assess the exact experimental conditions that generated them, from the short summary that can be presented in this article.

2) It might help with clarity for the reader if the results were displayed in a table so that comparisons can be easily made.

We have now added a table with a summary of results.

3) I feel the authors need to be clear on the mode of delivery of the different berries given. For example, in a lot of the studies cited about blackcurrant, these were done with encapsulated spray-dried blackcurrant which allows for an easy consumption of a high anthocyanin intake. The way the current manuscript is written, it communicates as if participants in these studies consumed the berries. Consuming over 200 mg of anthocyanins from fresh blackcurrant would involve a large amount of blackcurrant which might be difficult for some participants to consume. Therefore, encapsulation makes it easier for participants to consume this amount of anthocyanins and also allows for blinding with placebo capsules.

For each study, we reported the form of delivery the way it is indicated in the corresponding manuscript. For example, in the mentioned section on blackcurrants, seven studies are described, and the way of delivery was reported, respectively, as: “a single dose of blackcurrant extract drink”. “a daily dose of blackcurrant extract”, “600 mg blackcurrant extract”, “a daily dose of blackcurrant juice”, “a daily blackcurrant extract”, “two daily 300 mg capsules of blackcurrant extract”. We honestly fail to see how this would communicate as if participants consumed the fresh berries!

Anyway, in the table we have now clearly mentioned the use of freeze-dried powder when this is stated in the manuscripts (in several manuscripts the use of freeze-dried powder is not specifically mentioned, although it can be assumed).

4) Did any of the studies cited that examined blood pressure responses use a low polyphenolic diet before the laboratory visit (i.e. a washout diet before visiting the laboratory/ clinic)? A strength to many of the studies that didn’t use a washout diet, is that blood pressure changes are observed following supplementation of the berries on top of the participants normal diet – in turn, increasing ecological validity.

Very few studies mentioned a low-phenolic diet / washout period before treatment. We agree that this may indeed be relevant to the significance of findings, and we have now added the following observation about this to the discussion:

“With few exceptions, the study design did not involve a washout period before the start of the intervention, during which participants would be asked to follow a low-polyphenols diet, or to abstain from berry consumption. This means that berry treatment was in addition to the normal diet of the participants thus strengthening the ecological validity of those studies that report changes in blood pressure”.

5) A key consideration that the authors should also refer to is the role of anthocyanin metabolites. Due to the bioavailability of whole anthocyanins being low, it is likely that the metabolites and secondary phase metabolites are causing the blood pressure responses. Therefore, this content could be included within the discussion as it is an important consideration. Especially for the studies demonstrating changes following different doses and time-durations.

We have now included this consideration in the discussion:

“The metabolic fate of ACNs during digestion and following absorption is intricate and still largely unknown. The vast majority are not absorbed intact, but after hydrolysis and partial degradation to other phenolic compounds or following fermentation by the gut microbiota. Once in the bloodstream, whole ACNs, their parent compounds, degradation products and microbial metabolites, are all quickly converted to methyl, glucuronide, and sulfate conjugated metabolites by phase I and phase II enzymes. 

Thus, it is likely that most of the biological responses to ACN consumption is due to the activity of secondary phase metabolites, and that differences in ACN metabolism account for a large proportion of the observed variability in blood pressure results. This may involve not only individual differences in ACN metabolism, but also different metabolic responses related to different doses, frequency of intake, and time duration of interventions. This is a factor that will need to be kept in mind and more carefully investigated in future studies.”

Round 2

Reviewer 1 Report

The manuscript was revised well.

Author Response

We are glad that the reviewer is satisfied with the changes we have made.

Reviewer 3 Report

Thank you to the authors for responding to my original comments on their manuscript. I am happy with the justification/ explanations the authors give to my comments.

The manuscript is clearer and improved. The figure helps with explanation of mechanisms and the table helps demonstrate study details on the studies included within this review. I have a few small comments that should be addressed for accuracy.

Within table 1 ( reference 129), this study utilised a 7-day intake of blackcurrant and placebo cross over with randomisation, however the authors state it was a 2-week treatment duration. Therefore, this should be changed.

Within table 1 the authors state C for controlled. They also state PC for placebo controlled.  Therefore, could the authors clarify the difference between these points for clarity for the reader.

Author Response

We are glad that the reviewer’s doubts have now been clarified and that the changes we have made according to his/her suggestions are deemed satisfactory.

Within table 1 ( reference 129), this study utilised a 7-day intake of blackcurrant and placebo cross over with randomisation, however the authors state it was a 2-week treatment duration. Therefore, this should be changed.

We have now corrected the mistake.

Within table 1 the authors state C for controlled. They also state PC for placebo controlled.  Therefore, could the authors clarify the difference between these points for clarity for the reader.

We used “controlled” to indicate studies that had a control group, but not a placebo (for example, a berry drink versus water). We have now clarified the definition as “C, controlled without placebo”.